# Role of Sodium/Calcium Exchangers in Tumors

**DOI:** 10.3390/biom10091257

**Published:** 2020-08-31

**Authors:** Barbora Chovancova, Veronika Liskova, Petr Babula, Olga Krizanova

**Affiliations:** 1Institute of Clinical and Translational Research, Biomedical Research Center, Slovak Academy of Sciences, Dubravska cesta 9, 845 45 Bratislava, Slovakia; barbora.chovancova@savba.sk (B.C.); veronika.liskova@savba.sk (V.L.); 2Department of Physiology, Faculty of Medicine, Masaryk University, Kamenice 753/5, 625 00 Brno, Czech Republic; babula@med.muni.cz

**Keywords:** sodium-calcium exchanger, cancer cells, calcium, apoptosis

## Abstract

The sodium/calcium exchanger (NCX) is a unique calcium transport system, generally transporting calcium ions out of the cell in exchange for sodium ions. Nevertheless, under special conditions this transporter can also work in a reverse mode, in which direction of the ion transport is inverted—calcium ions are transported inside the cell and sodium ions are transported out of the cell. To date, three isoforms of the NCX have been identified and characterized in humans. Majority of information about the NCX function comes from isoform 1 (NCX1). Although knowledge about NCX function has evolved rapidly in recent years, little is known about these transport systems in cancer cells. This review aims to summarize current knowledge about NCX functions in individual types of cancer cells.

## 1. Background

Intracellular calcium ions are considered the most abundant secondary messengers in human cells, since they have a substantial diversity of roles in fundamental cellular physiology. Accumulating evidence has demonstrated that intracellular calcium homeostasis is altered in cancer cells and that this alteration is involved in tumor initiation, angiogenesis, progression and metastasis. Expression of several calcium transport systems has been shown to be altered in carcinogenesis (for review, see [1]).

The sodium/calcium exchanger (NCX) belongs to the calcium transport systems that control levels of intracellular calcium. The NCX was first identified and characterized by Baker et al. [2]. After almost twenty years, a crucial step forward was achieved by Philipson at al., who purified and cloned NCX1, a first isoform of this exchanger [3]. Afterwards, two other isoforms of the NCX—NCX2 and NCX3—were cloned and characterized [4,5]. Gene products of these three mammalian NCX isoforms (NCX1, NCX2, and NCX3) contain 930–970 residues, with ∼70% sequence overlay (for review, see [6]). The structure-based divergence of isoform/splice variants is a primary mechanism for regulatory diversification of eukaryotic NCXs. To date, at least 17 NCX1 and five NCX3 splice variants have been identified, although any splice variants have been found for NCX2. Expression of NCX splice variants is tissue-specific, e.g., the NCX1.1 variant is expressed in heart and skeletal muscle, whereas the NCX1.6 variant is expressed in the brain. Nevertheless, some tissues can express several NCX variants (e.g., kidneys express NCX1.2, NCX1.3, and NCX1.7 variants of NCX1). Functional and regulatory features of NCX splice variants are structurally predefined [6]. NCX1 is the only type of NCX that is ubiquitously expressed in almost every mammalian cell, e.g., in muscles, brain, kidneys, eyes, etc. [7,8]. Two other isoforms—NCX2 and NCX3—have been shown to be expressed predominantly in the brain (for review, see [9]), especially in the hippocampus [10] and the cerebellum [11], but also in the skeletal muscle [5]. Furthermore, in 2005, another isoform of the NCX, NCX4, was discovered. NCX4 occurs only in some fish species (e.g., fugu, tetraodon, zebrafish) [12] and apparently it plays a crucial role in a cardiovascular development [13]. All three isoforms have a bi-directional transport function, with the direction of Ca^2+^ movement through the NCX depending on the net electrochemical gradients for Na^+^ and Ca^2+^. In addition, the direction of transport depends on changes in membrane potential, whereby membrane depolarization augments Ca^2+^ influx and reduces Ca^2+^ efflux through the NCX and *vice versa* [14]. The main function of NCX1–3 isoforms is the transport of three sodium ions into the cell in exchange for one Ca^2+^ (forward mode). However, under special conditions, the NCX can also perform its function in an opposite way, i.e., promoting Ca^2+^ influx and Na^+^ efflux (reverse mode). Reverse mode of the NCX may be driven by sodium influx, thus triggering a massive secondary calcium elevation while promoting the export of sodium [15]. This mode of the NCX was shown to be predominant preferentially in pathological settings [16]. A functional comparative study by Linck et al. has shown that all three isoforms of the NCX display several similarities [17]. Currently, interest in the functional consequences of individual NCX isoforms and their modulation in various pathophysiological states is evolving. This review will focus on up-to-date insights into the role of the NCX in tumor progression, invasion, and its possible role in cancer treatment.

## 2. The NCX and Leukemia

Leukemia is a special type of cancer that affects the production and function of blood cells. Acute leukemia (AL) is a clonal malignant neoplastic disorder, which affects the whole population but is predominant in children. In children with AL, the serum calcium level was observed to be higher, compared to control age-matched group [18]. The natural steroid saponin, 3β,16β,17α-trihydroxycholest-5-en-22-one16-O-(2-O-4-methoxybenzoyl-β-d-xylopyranosyl)-(1-3)-2-O-acetyl-α-l-arabinopyranoside (OSW-1) isolated from *Ornithogalum saundersiae* is cytotoxic on a range of cancer cell lines (brain, lymphoma, ovarian, and pancreatic cell lines). In addition, a significant cytotoxicity has been demonstrated for leukemic cell lines, whereas the cytotoxic effect mediated by OSW-1 is marginal in normal lymphocytes. Garcia-Prieto et al. [19] have shown that the mechanism by which this compound increases intracellular calcium levels also includes the inhibition of the activity of sodium-calcium exchanger 1 (together with mitochondrial damage). OSW-1 caused pronounced elevation of cytosolic Ca^2+^ and reciprocally a simultaneous decrease in Na^+^ levels in human leukemia HL-60 cells in a time-dependent manner. These results are very similar to those found for isothiourea derivative 2-[2-[4-(4-nitrobenzyloxy)phenyl]ethyl]isothiourea methane sulfonate (KB-R7943), a well-known and potent NCX inhibitor that inhibits its reverse mode [19]. These results point to reverse mode NCX functioning in leukemia cells. However, possibility of utilizing NCX1 as a potential therapeutic target in leukemia is limited due to the essential role of this transporter in cardiac physiology [20]. Nevertheless, the functional consequences of NCX1 working in reverse mode in leukemia cells remain to be elucidated.

## 3. NCX1 in Solid Tumors

Importance of the NCX1 in carcinogenesis has been documented by several recent papers. Expression of the NCX1 is under the control of several transcription factors. Formisano et al. [21] summarized transcription factors that regulate expression of the *NCX1* gene. These transcription factors include cAMP response element-binding protein (CREB), hypoxia-inducible factor 1 (HIF-1), nuclear factor κB (NF-κB), and zinc finger transcription factor (Zn3-Sp1). Some of these factors are activated in tumor cells and thus can increase expression of the NCX1. After stimulation by oncogenes, tumor necrosis factor-α (TNF-α), or UV light, phosphorylation of the inhibitor of NF-κB occurs in a kinase cascade, followed by its degradation. Subsequently, NF-κB is released and translocated into the nucleus. This process continues with NF-κB binding to target DNA sequences (including that for NCX1) and regulates the transcription of several genes, which ensures adaptation to newly established conditions [22]. Since the majority of solid tumors became hypoxic when growing due to flawed angiogenesis, involvement of HIF-1 in transcriptional regulation of NCX1 is highly predictable. NCX1 brain promoter sequence was cloned and NCX1 was identified as a target gene of HIF-1. This fact favors its role in cell survival via the upregulation of the NCX1 transcript and protein [23]. Increased expression of the NCX1 under hypoxia was described also by Hudecova et al. [24], using HEK293 cells and mouse embryonic fibroblasts, both HIF-1-positive (MEF-HIF-1+/+) cells as well as HIF-1-deficient (MEF-HIF-1–/–) cells. This is in line with NCX1 upregulation due to ischemic insult in neuronal [9] and cardiac [8] tissues. In various types of tumors, NCX1 was shown to be upregulated due to hypoxic stimuli [25].

NCX1 expression has been found to be significantly higher in esophageal squamous cell carcinoma (ESCC) primary tissues compared to noncancerous ones. Overexpression of the NCX1 has also been found in tumor samples obtained from smoking patients. Comparison of human ESCC cell lines with non-tumor esophageal epithelial cell lines also revealed significantly higher protein levels of NCX1 [26]. The 4-(methylnitrosamino)-1-(3-pyridyl)-1-butanone (NNK), a tobacco-specific nitrosamine, potentiated calcium signaling via the reverse mode NCX. NNK supported proliferation and migration of human ESCC cells induced by NCX1 activation [27]. Xu et al. [28] have shown that the interaction among NHE1, NCX1, and calmodulin (CaM) regulates proliferation mediated by IL6 (a pro-inflammatory cytokine), migration, and invasion of human hepatocellular carcinoma (HCC) cells. These data point to the possible role of the NHE1, NCX1, and CaM molecular complex in tumor progression, and a novel signaling mechanism mediated by IL6 is proposed.

Transforming growth factor β (TGF-β) was also shown to upregulate NCX in megakaryocytes [29]. TGF-β plays an important role in the progression and processes of metastasis formation, as has been shown for HCC. In human HCC, the TRPC6/NCX1 (canonical transient receptor potential channel 6 and NCX1) complex promotes the effect of TGF-β on cell migration, invasion, and metastasis formation [30]. Formation of the TRPC6/NCX1 molecular complex is currently being studied for TGF-β-stimulated ([Ca^2+^]_cyt_) signaling in HCC cells [27]. In pancreatic cancer cells, TGF-β also induces entry of Ca^2+^ via TRPC1 and NCX1 and increases cytosolic levels of Ca^2+^. Calcium ions are essential for the activation of Ca^2+^-dependent protein kinase Cα (PKCα), with subsequent tumor cell invasion [31].

Upregulation of NCX1 by melatonin has been demonstrated in a variety of cancer cells [32,33]. Reduction of pancreatic damage by melatonin is mediated via the upregulation of NCX1 expression, which alleviates calcium overload in pancreatic acinar cells [33]. The mechanism of melatonin action is different in tumor cells compared to non-tumor endothelial cells, probably as a result of reorganization of signaling pathways in tumor cells [32]. In addition to the known effects of melatonin as a potent reactive oxygen species (ROS) scavenger [32,34], its ability to modulate intracellular calcium might significantly participate in decisions about apoptosis induction and the subsequent fate of tumor cells.

A special function has been attributed to NCX1 in breast cancer. Breast cancer is a group of tumors with different mechanism of induction, characteristics, vulnerability to treatment, and prognosis. Therefore, the modulation and function of NCX1 in different types of breast cancer may differ. Mahdi and colleagues [35] used estrogen receptor-positive breast cancer cell lines with the T47D and MCF-7 epithelial phenotype to study epithelial–mesenchymal transition (EMT) induced by TGF-β. TGF-β correlated with a high incidence of distant metastasis. It has been shown that TGF-β acted on the tumor cells and the surrounding stroma and promoted EMT, degradation of extracellular matrix, cell migration, and invasion. The authors showed that expression of the NCX1 decreased in TGF-βT47D cells, but not in TGF-β MCF-7 cells [35]. NCX1 silencing did not alter proliferation either in JIMT1 (a model of human epidermal growth factor receptor 2 (HER2)-positive breast cancer that possesses an amplification of HER2 receptor, but in which the cells do not respond to HER2 inhibition by trastuzumab, which selectively binds to the extracellular domain of HER2), or in MDA-MB-231 (a triple negative breast tumor-derived cell line). However, in the presence of nifedipine, silencing of the NCX1 did not alter JIMT1 cell proliferation, but significantly decreased the proliferation of MDA-MB-231. This fact suggests differing modulation of the NCX in individual breast cancer types [36].

In glioblastoma cells, the forward-mode NCX was shown to be predominant, since blocking the reverse mode NCX with selective inhibitors did not affect the tumor growth. It was shown that blocking the forward-mode NCX can suppress growth via Ca^2+^-mediated injury [37]. Furthermore, differently mutated melanoma cell lines (BRAF^V600E^-mutated and NRAS^Q61R^-mutated) present different Ca^2+^ homeostatic status and dependence on Ca^2+^ levels to survive and proliferate. Moreover, differences in the expression of NCX1 between these two types of cells result in differential susceptibility to NCX inhibitors [38].

## 4. In Solid Tumors NCX1 Operates Predominantly in Reverse Mode

Recently, the evidence that in solid tumors NCX1 acts in its reverse, “retrograde” mode, has become increasingly compelling. KB-R7943, the abovementioned blocker of the NCX1 reverse mode, is commonly used by several laboratories. Nevertheless, this blocker is not absolutely specific to the NCX; it also blocks N-methyl-D-aspartate receptor (NMDA) and inhibits mitochondrial respiratory complex I [39]. KB-R7943 depolarizes mitochondria independently of calcium (i.e., in a Ca^2+^-independent manner). KB-R7943 is also used as a reversible, activity-dependent blocker of the two most broadly expressed isoforms of ryanodine receptors, which are intracellular calcium channels localized on the endoplasmic reticulum. This mechanism contributes to its pharmacological and therapeutic activities [40]. Therefore, interpretation of the reverse-mode NCX cannot rely solely on proofs using this blocker. In prostate cancer cells, KB-R7943 was reported to inhibit all the processes connected with cell growth, cell cycle progression, and migration; and it has also been shown to induce apoptosis [41].

As already mentioned, the NCX operates in reverse mode in some cancer cells [41,42,43]. Recently it was shown that in hypoxic tumors NCX1 forms a metabolon with other proteins, which enhances the effectiveness of proton extrusion [25,43]. Solid tumors are highly metabolically active tissues. High metabolic activity leads to the formation of hypoxic regions that produce vast amounts of metabolic acids [44]. Thus, production of a bulk lactate, followed by acidification and decreasing pH value, is a common feature of tumor cells [45,46]. Malignant cells accumulate protons and intracellular pH becomes acidic. To prevent hyper-acidification of the intracellular space, activation of proton extruders is required. Among these systems, sodium/proton exchanger type 1 (NHE1) is an area of special interest [47]. The activation of NHE1′s inwardly-directed sodium gradient can drive the uphill extrusion of protons, which alkalinizes the intracellular environment and acidifies extracellular compartment matrix (for review, see [48]). A transport extrusion system is needed to extrude Na^+^ ions from the cells, in order to prevent overloading with Na^+^. This function is accomplished by the NCX1, working in reverse mode in cancer cells [25,43]. Taken together, in hypoxic cancer cells, NCX1 participates in the formation of a metabolon with several other proteins with the aim of keeping the intracellular pH slightly alkaline, while the extracellular pH becomes acidic. Presumably, NCX1 working in reverse mode may also participate in the calcification of solid tumors. Nevertheless, this hypothesis remains to be verified.

As was already discussed, the reverse-mode NCX might be driven by sodium influx, which can trigger a massive secondary calcium elevation, while promoting the export of sodium [15]. Therefore, in addition to NHE1, other sodium transport systems may also be altered during tumorigenesis. Promotion of the reverse-mode NCX could be achieved by suppressing Na^+^/K^+^-ATPase activity, which results in enhanced extracellular signal-regulated kinase 1/2 (ERK1/2) activation [49]. Indeed, myocardial ischemia disrupted the aerobic energy metabolism and decreased ATP levels, which resulted in compromising Na^+^/K^+^ ATPase function in the heart, resulting in the accumulation of Na^+^ in the cytosol, which altered the NCX to its reverse mode contributing to Ca^2+^ overload and consequent cell death [50]. The importance of Na^+^/K^+^-ATPase in carcinogenesis was described by Song et al. [51], who showed that in hepatocellular carcinoma, inhibitors of Na^+^/K^+^-ATPase (ouabain and digoxin) accelerated apoptosis and suppressed migration and cell growth. Furthermore, ouabain attenuated the stemness and chemoresistance of osteosarcoma cells [52]. Nevertheless, mutual crosstalk between the NCX and Na^+^/K^+^-ATPase remains to be further elucidated.

The 1-[2-(4-methoxyphenyl)2-[3-(4-methoxyphenyl)propoxy]ethyl-1H-imidazole hydrochloride (SKF 96365), a TRPC channel blocker, was experimentally used to suppress proliferation of glioblastoma cells. An inhibitory effect on Ca^2+^ entry via TRPC or TRPC6 channels is probably responsible for an anti-tumor effect [53]. SKF 96,365 was also shown to suppress growth of glioblastoma cells directly via enhancing the NCX operation in reverse mode and thus increasing intracellular Ca^2+^ concentration [54].

## 5. NCX2 and NCX3 in Solid Tumors

In contrast to NCX1, much less information is available about the possible involvement of NCX2 and NCX3 in carcinogenesis. Under physiological conditions, localization of the NCX3 isoform was described not only in brain and skeletal muscle, but was also detected in cells of the immune system and bone tissue. Interestingly, NCX3 transfected in baby kidney hamster cells was less vulnerable to chemical hypoxia compared to NCX1- and NCX2-transfected cells. This fact suggests that NCX3 probably plays a relevant protective role during chemical hypoxia [55].

Moreover, Liu et al. [56] reported that sub-cellular localization of NCX3 during the cell cycle is controlled by glycosylation. Cells can use a dynamic Ca^2+^ signaling toolkit that synchronizes the NCX3 changes in sub-cellular localization with the cell cycle. These authors observed that NCX3 localizes to the endoplasmic reticulum and the nuclear membrane during interphase, whereas during the S-phase, mitosis, and cell division, NCX3 is present in the plasma membrane. Surprisingly, the possible role of human NCX3 N-glycosylation in its translocation has not been fully elucidated. Modification of the NCX3 by N-linked glycosylation at a single asparagine residue (N45) is a basis for targeting this protein into the plasma membrane [56].

The activity of the NCX3 is probably regulated by PKC (regulated by diacylglycerol and/or Ca^2+^) and protein kinase A (PKA, regulated by cAMP). Although the forward mode of NCX3 was not affected by PKC stimulation, activity of the NCX3 working in reverse mode was significantly reduced upon PKC stimulation [57]. PKA was found to have similar effect to that of PKC. Interestingly, while both PKC and PKA alter the uptake of Ca^2+^ by NCX3 in the reverse mode, PKA on its own can affect the [Ca^2+^]i-stimulated forward exchange activity of two NCX3 variants that are the result of alternative splicing of the *SLC8A3* gene, NCX3-AC, and NCX3-B [57]. The authors suggested that this effect may be very important in the fight-or-flight response (acute stress) in skeletal muscle and long-term potentiation in the hippocampus. The pathophysiological relevance of sub-cellular NCX3 was suggested in Alzheimer’s disease, in which mitochondrial NCX3 is probably involved in Ca^2+^ replenishment of endoplasmic reticulum. By this mechanism it delays endoplasmic reticulum stress and cell death in the early phase of Alzheimer’s disease [58].

Pelzl et al. discovered that both, NCX3 transcript levels and activity of the exchanger are significantly higher in the cisplatin-sensitive human ovarian carcinoma cell line (A2780sens) compared to the cisplatin-resistant (A2780cis) one. The authors suggest that inhibition of Na^+^/Ca^2+^ exchanger may contribute to the development of better sensitivity to the chemotherapeutic agent cisplatin in ovary carcinoma cells [59]. The same team of investigators found that NCX3 is also overexpressed in UW228-3 medulloblastoma cells, resistant to therapy. The transcript level is also higher for K^+^-dependent Na^+^/Ca^2+^ exchanger 2 (NCKX2) and K^+^-dependent Na^+^/Ca^2+^ exchanger 5 (NCKX5), respectively. Similarly, silencing of the NCX3 enhanced apoptosis. This observation underscores a role of NCX3 in cell survival [60]. In vitro studies using melanoma cells with different metastatic capacities have shown that Ca^2+^ buffering capacity is lower in highly metastatic melanoma cells than in low metastasizing cells. Highly metastatic melanoma cells exhibited a sudden increase in intracellular Ca^2+^ levels, which was observed even in Na^+^-free medium, suggesting the contribution of the NCX in reverse mode to Ca^2+^ entry (for review, see [61]).

*SLC8A2* (solute carrier family 8, member 2 gene, encoding Na^+^/Ca^2+^ exchanger) might play a role in gliomas, the most common type of adult primary brain tumors, which have very low survival rates. Variable genetic aberrations are a typical molecular feature of most gliomas. They lead to inactivation of tumor suppressor genes, as well as amplification of oncogenes. Most frequently, genetic alteration (deletion) on the long arm of chromosome 19 is described. The *SLC8A2* gene is localized exactly on this spot. Based on this observation, it has been hypothesized that *SLC8A2* may be a possible tumor suppressor gene, and thus important for glioma development [62].

## 6. Conclusions

Despite the fact that NCX1 belongs to the best-studied antiporters and its role has been elucidated in a number of physiological and pathophysiological processes, information about its involvement in cancer cell survival and proliferation is quite sporadic. From the current literature it is apparent that in various tumors the NCX1 works in reverse mode and has a pro-survival role for cancer cells (Figure 1). However, its mechanism of action downstream from the NCX1 is still unknown. Nevertheless, only a small amount of information exists concerning the role of NCX2 and NCX3 in tumor fate. Downregulation of NCX2 based on DNA methylation in some tumors (e.g., gliomas) has been shown. Despite the fact that NCX3 protects neurons from injury and has very important neuroprotective functions, downregulation of NCX3 expression due to TNF-related apoptosis-inducing ligand (TRAL) has been elucidated. These facts indicate the involvement of NCX antiporters in processes connected with cell death and survival. They also indicate a possible pharmacological influence on their expression, for example through inhibitors of DNA methylation. However, it is necessary to distinguish between individual NCXs. A molecular biology approach—gene silencing or knockout of the gene—may produce better insights into the precise function of individual types of NCXs in tumors. In addition, the diversity of tumors should be taken into consideration. The absence of selective and potent inhibitors significantly impedes our knowledge about the functional consequences of individual NCX isoforms. To date, a huge effort has been expended in order to find the more potent and selective inhibitors acting on NCX isoforms in forward and reverse modes. These compounds might be of special interest in cancer biology, as they might form the basis of therapeutic tools.

## Figures and Tables

**Figure 1 biomolecules-10-01257-f001:**
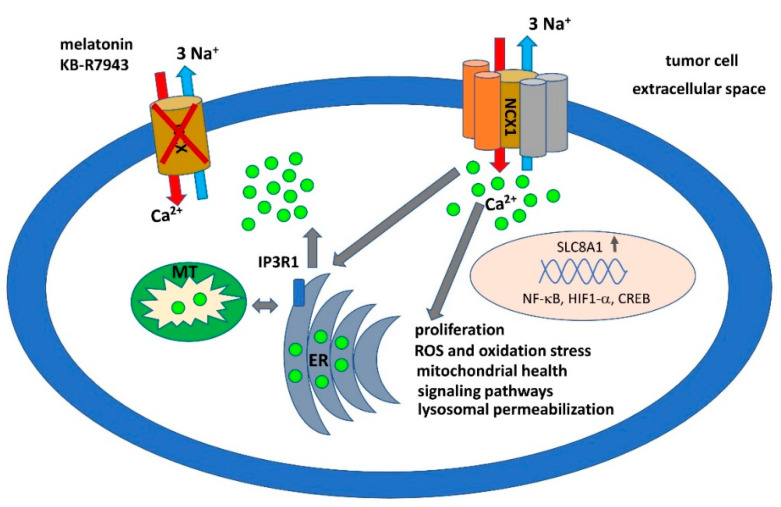
In tumor cells, expression of the gene encoding NCX1 (SLC8A1) is stimulated by transcription factors NF-κB, HIF-1α, or CREB. The NCX1 protein is incorporated into the protein complex in the membrane and acts preferentially in the reverse mode, which results in elevation of cytosolic calcium and activation of pro-survival metabolic pathways. Some natural (e.g., melatonin) and synthetic (e.g., KB-R7943) compounds are able to block NCX1′s transporting activity, which can disrupt the complex with other proteins and direct the cell to apoptosis.

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
