# Peer review of "Role of Sodium/Calcium Exchangers in Tumors"

_biomolecules, 2020, doi:10.3390/biom10091257_

Round 1
Reviewer 1 Report
Major concerns
The article by Chovancova and collaborators aims to give an up-to-date review on the role of Na+/Ca2+ exchangers in diverse tumors. The article covers well the literature regarding the exchangers and cancer; however, more information about the cellular context and the possible interplay between NCXs and other ion-transporting proteins is missed.
For instance, in this review it is mentioned that NCX1 works in reverse mode in solid tumor cells, which might be important as Na+-efflux pathway. Even though this is feasible under certain ion gradients, the Na+/K+ ATPase plays an essential role in Na+ efflux. I am wondering if there is information about Na+/K+ ATPase activity and protein levels in solid tumor cells. If so, I strongly recommend to include a small paragraph describing Na+/K+ ATPase activity in tumor cells compared to normal cells. Alternatively, some predictions about ATPase levels in solid tumor cells would significantly enrich this review.
Other major concerns:
Lines 33-34: the authors wrote that “under special conditions NCX can also perform in an opposite way….(reverse-mode). This sentence is quite vague; therefore, the authors should expand this statement with emphasis in the ionic conditions (Na+ and Ca2+ gradients) and the physiological and/or pathological condition that reverses the transport direction of NCX proteins.
Lines 179 and 186: this paragraph is hard to understand in the context of the role of NCX3 in solid tumor cells. Please re-write it.
Minor concerns
Line 28: replace “exprimed by expressed”
Line 48: it is important to mention that the cytotoxic effect mediated by OSW-1 is highly reduced in normal lymphocytes [Garcia-Prieto et al.].
Line 64: NF-κB is not well written. Make sure to correct this in the whole text.
Line 104: Same problem with TGF-β.
Line 143: Replace “alcalic by alkaline”
I recommend the acceptance of this article after addressing the concerns raised above.
Author Response
- More information about the cellular context and the possible interplay between NCXs and other ion-transporting proteins is missed. I am wondering if there is information about Na+/K+ ATPase activity and protein levels in solid tumor cells. If so, I strongly recommend to include a small paragraph describing Na+/K+ ATPase activity in tumor cells compared to normal cells. Alternatively, some predictions about ATPase levels in solid tumor cells would significantly enrich this review.
In fact, very little is known about the Na+/K+ ATPase in tumors. Actually, we have found just two papers showing that inhibitors of Na+/K+-ATPase accelerated apoptosis and suppressed migration and cell growth in hepatocellular carcinoma, or attenuated stemness and chemoresistance of osteosarcoma cells. Up to now, there is no paper about mutual interaction of the NCX and Na+/K+ ATPase.
To clarify this issue also in the manuscript, we added following paragraph at the end of section 4.
As it was already discussed, reverse mode of NCX might be driven by sodium influx, which can trigger a massive secondary calcium elevation while promoting export of sodium (Gerkau et al., 2018). Therefore, besides NHE1 also other sodium transport systems might be altered during tumorigenesis. Promotion of the reverse-mode NCX could be achieved also by suppressing Na+/K+-ATPase activity, which results in enhanced extracellular signal-regulated kinase 1/2 (ERK1/2) activation (Andrikopoulos et al., 2015). Indeed, myocardial ischemia disrupted the aerobic energy metabolism and decreased ATP levels, which resulted in compromising Na+/K+ ATPase function in the heart, resulting in accumulation Na+ in the cytosol, which alters the NCX to reverse mode contributing to Ca2+ overload and consequent cell death (Tani, 1990). Importance of Na+/K+-ATPase in carcinogenesis was described by Song and co-workers (2020), who showed that in hepatocellular carcinoma inhibitors of Na+/K+-ATPase (ouabain and digoxin) accelerated apoptosis and suppressed migration and cell growth. Also ouabain attenuated stemness and chemoresistance of osteosarcoma cells (Guo et al., 2019). Nevertheless, mutual crosstalk between NCX and Na+/K+-ATPase remains to be further elucidated.
- Lines 33-34: the authors wrote that “under special conditions NCX can also perform in an opposite way….(reverse-mode). This sentence is quite vague; therefore, the authors should expand this statement with emphasis in the ionic conditions (Na+ and Ca2+ gradients) and the physiological and/or pathological condition that reverses the transport direction of NCX proteins.
In line with this comment we rewrote corresponding part as follows (new information is shown in italics):
All three isoforms have bi-directional transport function, when direction of Ca2+ movement through the NCX depends on the net electrochemical gradients for Na+ and Ca2+. Also, the direction of transport depends on changes in membrane potential, whereby membrane depolarization augments Ca2+ influx and reduces Ca2+ efflux through the NCX and vice versa (Blaustein and Lederer, 1999).The main function of NCX1-3 isoforms is a transport of three sodium ions into the cell in exchange for one Ca2+ (forward mode). However, under special conditions NCX can also perform in an opposite way, i.e. promote Ca2+ influx and Na+ efflux (reverse mode). Reverse mode of NCX might be driven by sodium influx, thus triggering a massive secondary calcium elevation while promoting export of sodium (Gerkau et al., 2018). This mode of NCX was shown to be predominant preferentially in pathological settings (Khananshvili et al,. 2014).
- Lines 179 and 186: this paragraph is hard to understand in the context of the role of NCX3 in solid tumor cells. Please re-write it.
Above-mentioned paragraph was rewritten as follows:
Pelzl and his colleagues discovered that NCX3 transcript level and activity of the exchanger is significantly higher in therapy cisplatin-sensitive human ovarian carcinoma cell line (A2780sens) compared to therapy cisplatin-resistant (A2780cis) one. The authors suggest that inhibition of Na+/Ca2+ exchanger may participate to better sensitivity to chemotherapeutic agent cisplatin in ovary carcinoma cells (Pelzl et al., 2015).
- Line 28: replace “exprimed by expressed”
We apologize for this mistake, it was corrected.
- Line 48: it is important to mention that the cytotoxic effect mediated by OSW-1 is highly reduced in normal lymphocytes [Garcia-Prieto et al.].
We added this information into the text.
- Line 64: NF-κB is not well written. Make sure to correct this in the whole text. Line 104: Same problem with TGF-β.
We must apologize for not properly written symbols in the text, which happened after conversion of the text into the journal template. We went through the manuscript and corrected all symbols.
- Line 143: Replace “alcalic by alkaline”
This word was corrected.
Reviewer 2 Report
The review from Krizanova’s lab on the Na+/Ca2+ exchangers is well written and of interest as it focus on the role of these transporters in tumor cells, particularly when working in a reverse mode, an area little explored. There are some points that should be taken in consideration to improve the manuscript before be acceptable for publication.
Major points:
- The driving force responsible for the Na+/Ca2+ exchanger working in reverse (Na+ exit and Ca2+ entry) should be explained with some detail in the Introduction.
- Lines 55-56. The phrase “These results point to reverse mode NCX functioning in leukaemia cells and suggest a possibility for NCX1 as a potential therapeutic target in leukemia.” Should be underplayed. A concern is that inhibition of NCX1, even when working in a reverse mode, could have cardiotoxic effect due to the relevant role of this transporter in cardiac physiology (e.g. Shattock et al. 2015 J. Physiol. 593: 1361-1382; Levesque et al. 1991 Ann. N.Y. Acad. Sci. 639: 386-397).
- Lines 97-99. A reference should be given in the phrase “Besides the known effects of melatonin as a potent reactive oxygen species (ROS) scavenger…” (e.g. Tan et al. 2002 Curr. Top. Med. Chem. 2: 181-197).
- What is the difference between A2780sen and A2780cis ovarian carcinoma cell lines? (lines 180-181). It is important to understand the meaning of the phrase.
Minor points:
- Line 28. “exprimed” should read “expressed”.
- Lines 45-46. In the chemical name of OSW-1 some Greek symbols are missing: (3β,16β)-3,17-Dihydroxy-22-oxocholest-5-en-16-yl 2-O-acetyl-3-O-[2-O-(4-methoxybenzoyl)-β-D-xylopyranosyl]-α-L-arabinopyranoside.
- Line 46. The scientific name “Ornithogalum saundersiae” should be written in Italics.
- In many instances the Greek symbols in: nuclear factor-kappaB, protein kinase Calpha and transforming growth factor-beta are missing.
- Line 49 “that mechanism” should read “that the mechanism”.
- Superscripts should be used in + and 2+ where appropriate (e.g. Na+/Ca2+).
- Line 143. “alcalic” should read “alkaline”.
- Line 153. “On contrary to” should read “In contrast to”.
- Line 184. NCKX2 and NCKX5 should be defined: K+-dependent Na+/Ca2+ exchanger.
- Line 188. “very small survival” should read “very low survival”.
- Reference list: Please unify style avoiding the use of upper cases in the title of the articles, use superscripts in + and 2+ when referring to Na+ and Ca2+, and use Greek symbols where appropriate.
Author Response
- The driving force responsible for the Na+/Ca2+ exchanger working in reverse (Na+ exit and Ca2+ entry) should be explained with some detail in the Introduction.
As mentioned in the comment 2 of reviewer 1, corresponding part of Background was modified as follows (new information added to the Introduction is shown in italics):
All three isoforms have bi-directional transport function, when direction of Ca2+ movement through the NCX depends on the net electrochemical gradients for Na+ and Ca2+. Also, the direction of transport depends on changes in membrane potential, whereby membrane depolarization augments Ca2+ influx and reduces Ca2+ efflux through the NCX and vice versa (Blaustein and Lederer, 1999).The main function of NCX1-3 isoforms is a transport of three sodium ions into the cell in exchange for one Ca2+ (forward mode). However, under special conditions NCX can also perform in an opposite way, i.e. promote Ca2+ influx and Na+ efflux (reverse mode). Reverse mode of NCX might be driven by sodium influx, thus triggering a massive secondary calcium elevation while promoting export of sodium (Gerkau et al., 2018).This mode of NCX was shown to be predominant preferentially in pathological settings (Khananshvili et al,. 2014).
- Lines 55-56. The phrase “These results point to reverse mode NCX functioning in leukaemia cells and suggest a possibility for NCX1 as a potential therapeutic target in leukemia.” Should be underplayed. A concern is that inhibition of NCX1, even when working in a reverse mode, could have cardiotoxic effect due to the relevant role of this transporter in cardiac physiology (e.g. Shattock et al. 2015 J. Physiol. 593: 1361-1382; Levesque et al. 1991 Ann. N.Y. Acad. Sci. 639: 386-397).
We must agree with this comment and we modified the above-mentioned sentence as follows:
However, recently a possibility to utilize NCX1 as a potential therapeutic target in leukemia is limited due to the essential role of this transporter in cardiac physiology (Shattock et al., 2015).
- Lines 97-99. A reference should be given in the phrase “Besides the known effects of melatonin as a potent reactive oxygen species (ROS) scavenger…” (e.g. Tan et al. 2002 Curr. Top. Med. Chem. 2: 181-197).
As suggested by this reviewer, citations of Tan et al., 2002 and Chovancova et al., 2017 were added to above-mentioned sentence.
- What is the difference between A2780sen and A2780cis ovarian carcinoma cell lines? (lines 180-181). It is important to understand the meaning of the phrase.
A2780sens is cisplatin sensitive human ovarian carcinoma cell line and A2780cis is human ovarian carcinoma cell line, which is resistant to cisplatin. We modified the sentence in the manuscript as follows to clarify the difference in cell lines:
Pelzl and his colleagues discovered that NCX3 transcript level and activity of the exchanger is significantly higher in cisplatin-sensitive human ovarian carcinoma cell line (A2780sens) compared to cisplatin-resistant (A2780cis) one.
- Line 28. “exprimed” should read “expressed”.
We apologize for this mistake, it was corrected.
- Lines 45-46. In the chemical name of OSW-1 some Greek symbols are missing: (3β,16β)-3,17-Dihydroxy-22-oxocholest-5-en-16-yl 2-O-acetyl-3-O-[2-O-(4-methoxybenzoyl)-β-D-xylopyranosyl]-α-L-arabinopyranoside. Line 46. The scientific name “Ornithogalum saundersiae” should be written in Italics. In many instances the Greek symbols in: nuclear factor-kappaB, protein kinase Calpha and transforming growth factor-beta are missing. Superscripts should be used in + and 2+ where appropriate (e.g. Na+/Ca2+).
We must apologize for not properly written symbols and formatting in the text, which happened after conversion of the text into the journal template. We went through the manuscript and corrected all symbols, formatting of the manuscript and superscripts.
- Line 49 “that mechanism” should read “that the mechanism”. Line 143. “alcalic” should read “alkaline”. Line 153. “On contrary to” should read “In contrast to”.
All these words we corrected.
- Line 184. NCKX2 and NCKX5 should be defined: K+-dependent Na+/Ca2+ exchanger.
Both abbreviations were explained as suggested by this reviewer.
- Line 188. “very small survival” should read “very low survival”.
Phrase was corrected.
- Reference list: Please unify style avoiding the use of upper cases in the title of the articles, use superscripts in + and 2+ when referring to Na+ and Ca2+, and use Greek symbols where appropriate.
Text was edited accordingly.
Reviewer 3 Report
Here we have very interesting, clear, and timely well-tuned review, provided by Chovancova et al. this review article summarizes important discoveries regarding functional contributions of sodium-calcium exchangers (NCXs) in different types of cancer. In general, there is an increasing body of evidence for NCX proteins playing an important role in altered calcium signaling and homeostasis in different types of tumors. This makes NCX isoform/splice variants very attractive for selective and cell-specific pharmacological targeting of NCX variants expressed in distinct types of tumor cells.
This review article can help many scientists, working in different fields. A major benefit of this paper is that it provides very important information on ample contributions of NCXs into specific tumors. It is very important to realize that we are only at the beginning of this astonishing new field and obviously, the underlying molecular and cellular mechanisms remain to be discovered. In general, this review article is well written and can significantly contribute to common efforts for a better understanding of molecular and cellular mechanisms underlying the role of NCX isoform/splice variants in cancer cell function, regulation and development as well as for effective pharmacological targeting of altered homeostasis in cancer cells. Despite these benefits, there are some concerns, which deserve a full attention before granting the manuscript publication.
1) Background is extremely short and practically useless for average readers. To help the readers, it would be worth to mention that NCX1 and NCX3 isoforms form splice variants, whereas NCX2 does not generate splice variants. It is essential to explain that NCX isoform/splice variants are expressed in a tissue-specific manner, where their functional and regulatory features are structurally predefined (for update review, please see Cell Calcium, 2020, 85:102131. doi: 10.1016/j.ceca.2019.102131(. This didactic information is essential for making it clear that tissue-specific pharmacological targeting of NCX variants is a major strategic approach in the near future, in general. Certainly, the relevant issues must be seriously taken into account for forthcoming cancer pharmacology.
2) More clear definition of the forward and reverse modes is required in the text since these opposing transport modes could be crucial for cancer cell function and pharmacological targeting. The reason for this stems from the fact that altered relationships between these two transport modes may significantly shape Ca signaling and homeostasis in general and especially in tumor cells. As a minimum the authors must mention review articles describing the physiological and pathological conditions and mechanisms that can shift the ratio between the forward and reverse modes of NCX and thus, may control numerous cellular functions (e.g. see Physiol Rev, 1999, 79:763-854; Plügers Arch, 2014, 466:43–60). Once again, the relevant alterations could be characteristically specific in distinct tumor types and thus, could be extremely important for effective pharmacological targeting.
3) In general, the references are carefully chosen, albeit some recent articles, dealing with NCX contributions in cancer are missing (please see the list below). This reviewer urges the authors to consider a decent citation and brief reviewing of indicated papers in the sake of valuable scientific information and farness. If the authors think that some specific publications are inappropriate for reference, they must kindly explain the reasons.
Reference list for completion
1) Targeting calcium signaling in cancer therapy.
Cui C, Merritt R, Fu L, Pan Z.Cui C, et al.
Acta Pharm Sin B. 2017 Jan;7(1):3-17. doi: 10.1016/j.apsb.2016.11.001.
2) BRAF and NRAS mutated melanoma: Different Ca2+ responses, Na+/Ca2+ exchanger expression, and sensitivity to inhibitors
Gabriela Nohemi Nuñez Estevesa, Letícia Silva Ferraza, Marcela Maciel Palacio Alvarezb, Claudia Alves da Costac, Rayssa de Mello Lopesc, Ivarne Luis dos Santos Tersariolb, Tiago Rodriguesa, Cell Calcium 90 (2020) 102241
3) Plasma membrane Ca2+-permeable channels and sodium/calcium exchangers in tumorigenesis and tumor development of the upper gastrointestinal tract.
Ding J, Jin Z, Yang X, Lou J, Shan W, Hu Y, Du Q, Liao Q, Xu J, Xie R. Ding J, et al.
Cancer Lett. 2020 Apr 10;475:14-21. doi: 10.1016/j.canlet.2020.01.026.
4) Na+/Ca2+ exchangers: Unexploited opportunities for cancer therapy?
Rodrigues T, Estevez GNN, Tersariol ILDS.Rodrigues T, et al.
Biochem Pharmacol. 2019 May;163:357-361. doi: 10.1016/j.bcp.2019.02.032.
5) Blockade of the forward Na+ /Ca2+ exchanger suppresses the growth of glioblastoma cells through Ca2+ -mediated cell death.
Hu HJ, Wang SS, Wang YX, Liu Y, Feng XM, Shen Y, Zhu L, Chen HZ, Song M.Hu HJ, et al.
Br J Pharmacol. 2019 Aug;176(15):2691-2707. doi: 10.1111/bph.14692. Epub 2019 Jun 17
Author Response
1) Background is extremely short and practically useless for average readers. To help the readers, it would be worth to mention that NCX1 and NCX3 isoforms form splice variants, whereas NCX2 does not generate splice variants. It is essential to explain that NCX isoform/splice variants are expressed in a tissue-specific manner, where their functional and regulatory features are structurally predefined (for update review, please see Cell Calcium, 2020, 85:102131. doi: 10.1016/j.ceca.2019. 10213. This didactic information is essential for making it clear that tissue-specific pharmacological targeting of NCX variants is a major strategic approach in the near future, in general. Certainly, the relevant issues must be seriously taken into account for forthcoming cancer pharmacology.
We are grateful for this comment and we added following paragraph into the section Background:
Gene products of these three mammalian NCX isoforms (NCX1, NCX2, and NCX3) contain 930–970 residues with ∼70 % sequence overlay (for review see Khananshvili, 2020). The structure based divergence of isoform/splice variants is a primary mechanism for regulatory diversification of eukaryotic NCXs. Up to now, at least 17 NCX1 and 5 NCX3 splice variants were identified, albeit any splice variant was found for NCX2. Expression of NCX splice variants is tissue-specific, e.g. NCX1.1 variant is expressed in heart and skeletal musle, while NCX1.6 variant is expressed in brain. Nevertheless, some tissues can express several NCX variants (e.g. kidney expresses NCX1.2, NCX1,3 and NCX1.7 variants of NCX1). Functional and regulatory features of NCX splice variants are structurally predefined (Khananshvili, 2020).
2) More clear definition of the forward and reverse modes is required in the text since these opposing transport modes could be crucial for cancer cell function and pharmacological targeting. The reason for this stems from the fact that altered relationships between these two transport modes may significantly shape Ca signaling and homeostasis in general and especially in tumor cells. As a minimum the authors must mention review articles describing the physiological and pathological conditions and mechanisms that can shift the ratio between the forward and reverse modes of NCX and thus, may control numerous cellular functions (e.g. see Physiol Rev, 1999, 79:763-854; Plügers Arch, 2014, 466:43–60). Once again, the relevant alterations could be characteristically specific in distinct tumor types and thus, could be extremely important for effective pharmacological targeting.
This comment was the same as in the review 1. As mentioned in the comment 2 of reviewer 1, corresponding part of Background was modified as follows (new information added to the Background is shown in italics):
All three isoforms have bi-directional transport function, when direction of Ca2+ movement through the NCX depends on the net electrochemical gradients for Na+ and Ca2+. Also, the direction of transport depends on changes in membrane potential, whereby membrane depolarization augments Ca2+ influx and reduces Ca2+ efflux through the NCX and vice versa (Blaustein and Lederer, 1999).The main function of NCX1-3 isoforms is a transport of three sodium ions into the cell in exchange for one Ca2+ (forward mode). However, under special conditions NCX can also perform in an opposite way, i.e. promote Ca2+ influx and Na+ efflux (reverse mode). Reverse mode of NCX might be driven by sodium influx, thus triggering a massive secondary calcium elevation while promoting export of sodium (Gerkau et al., 2018). This mode of NCX was shown to be predominant preferentially in pathological settings (Khananshvili 2014).
3) In general, the references are carefully chosen, albeit some recent articles, dealing with NCX contributions in cancer are missing (please see the list below). This reviewer urges the authors to consider a decent citation and brief reviewing of indicated papers in the sake of valuable scientific information and farness. If the authors think that some specific publications are inappropriate for reference, they must kindly explain the reasons.
We are especially grateful to this reviewer for suggesting some recent citations and we apologize that we missed them. We read all of them carefully and included some relevant information into the manuscript (shown under each citation).
1) Targeting calcium signaling in cancer therapy.
Cui C, Merritt R, Fu L, Pan Z.Cui C, et al.
Acta Pharm Sin B. 2017 Jan;7(1):3-17. doi: 10.1016/j.apsb.2016.11.001.
Intracellular calcium ions are considered as the most abundant second messengers in human cells, since they have a substantial diversity of roles in fundamental cellular physiology. Accumulating evidences have demonstrated that intracellular calcium homeostasis is altered in cancer cells and the alteration is involved in tumor initiation, angiogenesis, progression and metastasis. Expression of several calcium transport systems has been shown to be altered in carcinogenesis (for review see Cui et al., 2017).
2) BRAF and NRAS mutated melanoma: Different Ca2+ responses, Na+/Ca2+ exchanger expression, and sensitivity to inhibitors
Gabriela Nohemi Nuñez Esteves, Letícia Silva Ferraz, Marcela Maciel Palacio Alvarez, Claudia Alves da Costa, Rayssa de Mello Lopes, Ivarne Luis dos Santos Tersariol, Tiago Rodrigues, Cell Calcium 90 (2020) 102241
Also, differently mutated melanoma cell lines (BRAFV600E mutated and NRASQ61R mutated) present different Ca2+ homeostatic status and dependence on Ca2+ levels to survive and proliferate. Moreover, differences in the expression of NCX1 between these two types of cells resulted in differential susceptibility to NCX inhibitors (Estevez et al., 2020).
3) Plasma membrane Ca2+-permeable channels and sodium/calcium exchangers in tumorigenesis and tumor development of the upper gastrointestinal tract.
Ding J, Jin Z, Yang X, Lou J, Shan W, Hu Y, Du Q, Liao Q, Xu J, Xie R. Ding J, et al.
Cancer Lett. 2020 Apr 10;475:14-21. doi: 10.1016/j.canlet.2020.01.026.
The 4-(methylnitrosamino)-1-(3-pyridyl)- 1-butanone (NNK), a tobacco-specific nitrosamine, potentiated the calcium signaling via the reverse mode NCX. NNK supported proliferation and migration of human ESCC cells induced by NCX1 activation (Ding et al., 2020).
Formation of TRPC6/NCX1 molecular complex is currently studied for TGF-β-stimulated [Ca2+]cyt signaling in HCC cells (Ding et al., 2020).
4) Na+/Ca2+ exchangers: Unexploited opportunities for cancer therapy?
Rodrigues T, Estevez GNN, Tersariol ILDS.Rodrigues T, et al.
Biochem Pharmacol. 2019 May;163:357-361. doi: 10.1016/j.bcp.2019.02.032.
In vitro studies using melanoma cells with different metastatic capacity have shown that Ca2+ buffering capacity is lower in highly metastatic melanoma cells than in low metastasing cells. Highly metastatic melanoma cells exhibited a sudden increase in intracellular Ca2+ levels, which was observed even in Na+-free medium, suggesting the contribution of NCX in the reverse mode to the Ca2+ entry (for review see Rodrigues et al., 2019).
5) Blockade of the forward Na+ /Ca2+ exchanger suppresses the growth of glioblastoma cells through Ca2+ -mediated cell death.
Hu HJ, Wang SS, Wang YX, Liu Y, Feng XM, Shen Y, Zhu L, Chen HZ, Song M.Hu HJ, et al.
Br J Pharmacol. 2019 Aug;176(15):2691-2707. doi: 10.1111/bph.14692. Epub 2019 Jun 17
In glioblastoma cells, forward mode NCX was shown to be predominant, since blocking the reverse mode of NCX with selective inhibitors did not affect the tumor growth. It was shown that blocking the forward NCX can suppress growth via Ca2+‐mediated injury (Hu et al., 2019).
Round 2
Reviewer 1 Report
The revised version of the Manuscript is significantly improved. The authors have included all the suggestions.
Consequently, I recommend its acceptance in the current form.
Reviewer 2 Report
The authors correctly made the changes requested.